# Dependence of Wind-Farm-Induced Gravity Waves and Wind-Farm Performance on Non-Dimensional Atmospheric Parameters and Simulation Configuration

Mehtab Ahmed Khan<sup>1</sup>, Matthew J. Churchfield<sup>2</sup>, and Simon J. Watson<sup>1</sup>

<sup>1</sup>Delft University of Technology, Netherlands

**Correspondence:** Mehtab Ahmed Khan (m.a.khan-2@tudelft.nl)

Abstract. This large eddy simulation (LES) study examines how wind-farm-induced atmospheric gravity waves (AGWs) and wind farm performance depend on non-dimensional atmospheric parameters and simulation configuration. A hypothetical aligned wind farm of actuator disks is simulated under neutral surface conditions, with a stable capping inversion and a mildly stable free atmosphere, to assess the effects of stratification beyond the atmospheric boundary layer (ABL) on ABL flow. Simulation setups fully resolving AGWs are validated to minimize spurious wave generation and reflection from the domain boundaries. The validated setup is then used to analyze AGW types and characteristics, as well as stratification impacts under conventionally neutral boundary layer (CNBL) conditions. These conditions are governed by four non-dimensional parameters: the Froude numbers of the free atmosphere and capping inversion (Fr,  $Fr_i$ ), and the aspect ratios of the ABL and wind farm ( $\tilde{H}_i$ ,  $S_h$ ).

Simulation configurations that fully resolve AGWs—capturing at least one wavelength both horizontally and vertically—yield the most realistic stratification effects on ABL flow, whereas partial or unresolved configurations produce nonphysical, channel-like behavior. A coherent description of the AGW phenomena is provided, highlighting the central role of capping inversion displacement in linking ABL fluctuations with AGWs. Trapped waves are confined within the capping inversion, while interfacial and internal waves aloft are identified as the AGW types most relevant to wind farm performance. The wavy inversion, analogous to an interfacial wave, forms converging and diverging zones that drive power fluctuations across the farm. The interfacial wavelength, measured over the wind farm, corresponds to one diverging, one converging, and one mildly diverging zone. As the interfacial wavelength decreases with  $Fr_i$ , multiple convergence—divergence zones develop under subcritical conditions ( $Fr_i < 1.0$ ), while for supercritical conditions ( $Fr_i > 1.0$ ), the wavelength approaches the farm length. Wave amplitude increases with decreasing  $\tilde{H}_i$  (i.e. shallower capping inversions).

Wind farm performance is most sensitive to  $\tilde{H}_i$ : shallow inversions increase blockage and reduce efficiency, while deeper layers enhance wake recovery. Increasing Fr,  $Fr_i$ , and  $S_h$  mitigates blockage and improves efficiency. Although local power fluctuations arise from AGWs, overall wind farm efficiency remains nearly constant with Fr and  $Fr_i$ , improving primarily with larger  $\tilde{H}_i$  and  $S_h$ .

<sup>&</sup>lt;sup>2</sup>National Renewable Energy Laboratory, USA

Preprint. Discussion started: 19 November 2025

© Author(s) 2025. CC BY 4.0 License.




Copyright statement. This work was authored in part by the National Renewable Energy Laboratory, operated by Alliance for Sustainable Energy, LLC, for the U.S. Department of Energy (DOE) under Contract No. DE-AC36-08GO28308. Funding provided by the U.S. Department of Energy Office of Energy Efficiency and Renewable Energy Wind Energy Technologies Office. The views expressed in the article do not necessarily represent the views of the DOE or the U.S. Government. The U.S. Government retains and the publisher, by accepting the article for publication, acknowledges that the U.S. Government retains a nonexclusive, paid-up, irrevocable, worldwide license to publish or reproduce the published form of this work, or allow others to do so, for U.S. Government purposes.

#### 30 1 Introduction

Numerical flow simulations are often used to investigate the interaction of wind farms with the atmosphere. Thermal stratification within the atmospheric boundary layer (ABL), which significantly affects the strength and structure of ABL turbulence, is often a focus of numerical studies. For example, many studies look to correlate wind turbine wake recovery with ABL stratification. The community often focuses solely on the thermal stratification *within* the ABL; however, it has become clear that the stratification *above* the ABL is also significant. A conventionally neutral boundary layer (CNBL), for instance, consists of a neutrally-stratified ABL capped by a strongly stable inversion layer and a weakly stable free atmosphere aloft. A wind farm operating in a CNBL can trigger atmospheric gravity waves (AGWs) in the layers above the ABL that contribute to the wind-farm blockage effect (Lanzilao and Meyers, 2023). For numerical models to accurately capture AGW effects, thermal and buoyancy effects must be well modeled and the simulation domain has to be properly configured. This study seeks to investigate AGW effects on wind farms while providing guidance on proper simulation design and configuration.

Researchers conduct simulations of offshore wind farms within the neutral ABL at varying levels of complexity depending on the research question. For example, if the focus is on turbulent wake effects, then simulating a truly neutral boundary layer (TNBL) with neutral stability throughout the entire vertical extent of the computational domain, or by neglecting buoyancy altogether, may be sufficient. To provide the next level of realism and complexity, one may include a capping inversion (a roughly 100-m thick layer of very stable stratification) followed by only a few hundred meters of the more weakly stably stratified free atmosphere above (Churchfield et al., 2012; Archer et al., 2013). With this additional realism, the capping inversion acts as a flexible lid on the top of the ABL that can move vertically in response to a strong perturbation from below, such as a large wind farm or terrain. The most realistic simulation configuration, though, is to simulate a CNBL that is capable of fully resolving AGWs by including a capping inversion followed by many kilometers of the weakly stably stratified free atmosphere above. Such a simulation configuration allows the study of wind farm behavior as a function of both ABL stratification and AGW characteristics, a subject that is rapidly gaining attention (Lanzilao and Meyers, 2024). Finally, a recent simplified approach is to neglect any stratified layers and to place a stress-free, rigid, impermeable boundary condition at the expected height of the capping inversion, instead of well above it. Using such a top boundary condition at the top of the computational domain is commonplace, but to use it as an inflexible capping inversion is uncommon (Stipa et al., 2024). Although this simplified simulation configuration has similarities with the other configurations, it will contain inaccuracies in the velocity and pressure fields because the ABL height cannot adjust to obstacles such as wind turbines (Smith, 2023).

Preprint. Discussion started: 19 November 2025

© Author(s) 2025. CC BY 4.0 License.



Properly defining the height of the top boundary of the simulation domain is always critical. Suppose one performs a wind farm flow simulation and uses an impermeable top boundary condition; not allowing enough distance between the top of the ABL and the top boundary will unrealistically constrain the flow within the ABL leading to errors. The top boundary should be placed high enough that the boundary layer can freely expand in the vertical direction while its flow advects through the wind farm, as shown by Stevens et al. (2014), for example. If one wants to capture AGW effects, the simulation domain should be tall enough to accommodate at least one AGW wavelength, usually extending a few kilometers above the capping inversion, as shown by Khan et al. (2024b, a). The impact of using a domain height shorter than one AGW vertical wavelength is unclear. In addition to the location of simulation boundaries, the ability of those boundaries to properly handle AGW reflections is important. If an AGW reflects off of simulation boundaries, it can seriously contaminate the solution. Khan et al. (2024a) explains the criterion that if 90% or more of the upward-propagating wave energy is absorbed or let out of the domain, the solution will be sufficiently reflection-free. The impact of AGW reflections on predicted wind farm power output has not been thoroughly explored.

If the wind farm is large enough and contains tall enough turbines relative to the ABL height, AGWs are inevitable under CNBL conditions. To simulate this situation, the CNBL configuration with a many-kilometers-tall domain is the most realistic amongst those discussed above, especially for offshore sites (Khan et al., 2024a; Lanzilao and Meyers, 2024). The literature focuses on the impacts of AGWs on wind farm performance, with limited insight into the AGW phenomenon itself (Allaerts et al., 2018; Lanzilao and Meyers, 2024; Stipa et al., 2024). For instance, the literature consistently shows that AGWs cause a feedback effect on the flow in the ABL. Smith (2010); Allaerts and Meyers (2015); Lanzilao and Meyers (2024); Stipa et al. (2023, 2024) argue that AGWs are responsible for the adverse pressure gradients upstream of the wind farm that create a blockage effect and favorable pressure gradients over and downstream of it. Smith (2010) describes the capping inversion displacement caused by the deceleration of the flow within the wind farm as the trigger of the AGWs. Stipa et al. (2024) describes how the capping inversion displacement couples AGW flow perturbations in the free atmosphere to perturbations within the ABL.

A good description of how AGWs then affect the capping inversion displacement, which trigger those waves in the first place, is lacking. Moreover, different types of wind farm-induced AGW are mentioned in the literature without clearly distinguishing them. For example, the work of Khan et al. (2024a, b) is related to "internal" AGWs, whereas Smith (2010); Allaerts and Meyers (2015); Lanzilao and Meyers (2024); Stipa et al. (2024) speak of "interfacial" AGWs as most critical for wind farm performance. "Trapped" AGWs in the capping inversion and even AGWs within the ABL are mentioned (Khan et al., 2024b; Allaerts and Meyers, 2017; Lanzilao and Meyers, 2024). There is a need to describe the AGW phenomena clearly, distinguish all the kinds of AGWs, and identify how each kind influences wind farm performance and simulation set-up.

Past studies of AGW impacts on wind farms focused on either dimensional and non-dimensional quantities. Allaerts and Meyers (2018) and Lanzilao and Meyers (2024) focused on dimensional quantities of interest including ABL depth and thermal stratification strength of the capping inversion and free atmosphere. In contrast, Smith (2010) focuses on non-dimensional quantities. Similarly, Khan et al. (2024a, b) examined various non-dimensional quantities used to establish an optimal large-eddy simulation (LES) setup for capturing realistic AGW behavior. In this work, we adopt this non-dimensional approach to

Preprint. Discussion started: 19 November 2025

© Author(s) 2025. CC BY 4.0 License.


investigate AGWs and their impact on wind farm performance in a CNBL. We have three objectives, all related to realistic and accurate modeling and understanding of wind-farm-atmosphere interactions under offshore CNBL conditions:

- We characterize the effects of modeling choices on the simulation of wind-farm-atmosphere interaction, by comparing four configurations: a) the TNBL, b) the CNBL with a tall domain height, c) a CNBL with a neutrally stratified free atmosphere, and d) replacing the flexible capping inversion with an impermeable, rigid top boundary.
- We examine the impact of AGW reflections on wind farm power output.
- We study the dependence of AGWs and wind farm performance on various non-dimensional parameters defined by Khan et al. (2024b).

The remainder of this article is organized as follows: Section 2 explains the different kinds of AGWs observed in wind-farm-atmosphere interactions. Section 3 summarizes the non-dimensional metrics used to analyze AGW characteristics and wind farm flow dynamics. The models and simulation configurations are also explained in this section. We compare the different approaches in simulating neutral ABL conditions in Section 4.1. Section 4.2 shows the impact of AGW reflections on wind farm power output in a CNBL. Section 4.3 and Section 4.4 show the dependence of AGW characteristics and wind farm performance on the non-dimensional parameters, respectively. We conclude with Section 5.

#### 2 Different Kinds of Wind Farm-Induced AGWs

Various kinds of AGWs are described in the literature, and we sometimes find it challenging to synthesize all the information. Let us give some examples: "interfacial" and "internal" AGWs are described as occurring within the capping inversion and in the free atmosphere, respectively (Khan et al., 2024a; Smith, 2010; Allaerts and Meyers, 2017; Lanzilao and Meyers, 2024). Allaerts and Meyers (2017) and Lanzilao and Meyers (2024) describe interfacial waves as two-dimensional waves somehow bound to the top of the capping inversion. They also discuss waves similar to interfacial waves inside the neutral boundary layer, reaching all the way down to the hub height. Khan et al. (2024b) identify "trapped" waves confined within the capping inversion, and Ollier and Watson (2023) describes topographically-induced "trapped lee" waves affecting wind farms. Lee waves may also be induced by wind farms, which can affect the flow in a way similar to shallow topography. Because the literature is vast and spread out, we attempt to briefly weave the different kinds of AGWs and their effects into an essential story, giving the reader enough background to understand the results and discussion presented in this article.

The causes of wind-farm-induced atmospheric gravity waves involve mass conservation, buoyancy, and wave theory. A wind farm is a sink of horizontal momentum and appears similar to a porous blockage, causing the flow to decelerate through the farm. It then begins to accelerate behind the farm as wind farm wake recovers. As a result of flow continuity, the flow in the ABL is deflected upward at the start of the wind farm and downward at the end of it in response to the horizontal deceleration and acceleration (Allaerts and Meyers, 2017). The capping inversion, as its name implies, is the "cap" on top of the ABL. It is a relatively thin layer of air that is strongly stable in thermal stratification, so it acts as a flexible cap or lid atop the boundary layer. Vertical motions of turbulence within the boundary layer generally do not penetrate this layer—it strongly suppresses


Figure 1. Contours of vertical velocity, w, on a vertical streamwise slice through the middle of the wind farm for Froude number of 0.15, inversion Froude number of 0.75, and wind farm shape factor of 0.025: (a) temporally averaged field showing the internal and interfacial waves; (b) temporally averaged field showing the inversion layer displacement, interfacial waves, and their signature in the ABL; and (c) instantaneous field from a non-precursor driven simulation that shows the trapped waves in the inversion layer triggered when the wind farm-induced turbulence hits the inversion layer. The inversion layer mid-point height is at z = 500 m, and the wind farm start and end are at x/L = -0.5 and +0.5, respectively, where L is wind farm length.

them with buoyancy forces. Therefore, it displaces upward over the wind farm and downwards at the end of it due to the flow deflected by the wind farm in that manner. To the flow aloft in the free atmosphere, this deflected capping inversion appears similar to a gentle hill above the wind farm. As does the vertical flow caused by real hills, the vertical flow over the displaced capping inversion activates buoyancy forces that tend to restore air displacements in the free atmosphere and the inversion layer itself, resulting in AGWs.

AGWs can be classified into a few different types based on their characteristics and whether they occur in the inversion layer, in the free atmosphere, or at the interface of the two. AGWs are composed of an entire spectrum of wavelengths; however, there is an apparent wavelength dictated by local thermal stratification strength and horizontal wind speed. AGWs propagating in the free atmosphere are referred to as "internal" gravity waves, which are steady if the conditions are steady, as shown in Fig. 1(a). Internal waves have a tendency to curve—like an arc of a circle—downstream of the obstacle causing them, and may mildly perturb the inversion layer, but only downstream of the obstacle. AGWs that may occur in the capping inversion, if its depth is great enough, are referred to as "trapped" waves because they are confined within that layer. The unsteady turbulence that is constantly pushing on the capping inversion from below contributes to the unsteadiness of these trapped waves. An

Preprint. Discussion started: 19 November 2025

© Author(s) 2025. CC BY 4.0 License.







example of trapped waves can be seen in the instantaneous vertical velocity field in Fig. 1(c). Because the free atmosphere and capping inversion have different stratification strengths and wind speeds, they support different apparent wavelengths of AGWs. As shown in Fig. 1(b), at the interface between the free atmosphere and the capping inversion, we see "interfacial" waves that smoothly blend the internal and trapped gravity waves, and they have their own apparent wavelength. The wavelengths of trapped waves not supported in the free atmosphere are called evanescent waves, and they decay with height. They are contained and advected along the interface (Sachsperger et al., 2015). For subcritical conditions—wave speed faster than background wind speed—a large part of the wave spectrum becomes evanescent and travels faster than the background wind speed along the interface, especially downstream. These evanescent waves downstream of the obstacle are the "trapped lee waves", which can relatively strongly perturb the capping inversion.

Although we summarize AGW formation as a succession of events, we stress that it is really a tightly coupled integrated set of phenomena that happen almost in lockstep. If one parameter in the system changes, the entire ABL-AGW system quickly responds, settling toward a new equilibrium.

#### 3 Methods

#### 3.1 Simulation configuration based on Non-dimensional Parameters

We simulate a wind farm consisting of a regular  $10 \times 5$  array of wind turbines with a spacing of five rotor diameters in the streamwise and three rotor diameters in the lateral direction. We use the Toolbox fOr Stratified Convective Atmospheres (TOSCA) (Stipa et al., 2023), an LES code specifically developed to simulate atmospheric flows through wind farms. TOSCA can simulate turbines as actuator lines or actuator discs with controllers for blade pitch, nacelle yaw, and generator speed. This study uses the uniform actuator disc model with a thrust coefficient, with normalizing velocity taken at the disc,  $C'_t$  of 1.3. The actuator discs have the same diameter and hub heights as those of the NREL-5MW wind turbine, 126 m and 90 m, respectively. TOSCA solves the filtered incompressible Navier-Stokes equations under non-hydrostatic conditions using the Boussinesq approximation for buoyancy. The code solves equations for continuity, momentum, and potential temperature. An advection damping layer and Rayleigh damping layers (RDL) are applied as body forces in the respective forcing zones through the momentum equation, details of which are given in Khan et al. (2024b) and Lanzilao and Meyers (2023). We denote the inlet, top, and outlet boundary RDLs as I-RDL, T-RDL, and O-RDL, respectively. Subgrid-scale turbulence is modeled using the dynamic Smagorinsky model with Lagrangian averaging (Meneveau et al., 1996). Further details are given by Stipa et al. (2023).

Figure 2 shows the simulation domain for the wind farm. Turbulent inflow conditions, including wind shear and veer, generated with offline precursor simulation are driven into the domain using an inflow Dirichlet boundary condition. Flow exits the domain using an outflow Neumann boundary condition. A typical wall shear stress lower boundary condition is used. It uses Monin-Obukhov similarity to relate the computed flow within the domain to the wall shear stress; Monin-Obukhov similarity is applied locally. An aerodynamic roughness height of 0.0001 m is used to mimic offshore conditions. Periodic boundary conditions are used in the transverse direction only. The upper boundary is set to a free-slip condition for wind speed.




**Figure 2.** Lateral view of the simulation domain that resolves ABL, the capping inversion, and the free atmosphere around a wind farm. Rayleigh damping layers are at the inlet, top, and outlet boundaries, and an advection damping layer is applied, overlapping the inlet Rayleigh damping layer, all in the free atmosphere. A CNBL potential temperature profile is shown on the right for completeness.

170 The temperature in the ABL is constant with height, then strongly and linearly increasing in the capping inversion, and finally linearly but more weakly increasing in the free atmosphere. This gives a buoyancy and Brunt-Väisälä frequencies constant with height in both layers. There is no heat flux at the ground, which maintains a neutral ABL throughout the simulation. We impose Coriolis forces corresponding to 41° north latitude.

The precursor simulations have the same configuration except that the inflow-outflow boundary conditions are replaced with periodic boundary conditions. The precursor spin-up time is 25 ks, and inflow data are saved for two flow-through times of the successor domain beyond the spin-up. A driving horizontal pressure-gradient controller maintains hub height velocity at 11.5 ms<sup>-1</sup> with the direction aligned to the horizontal axis. The computed pressure-gradient is also saved and used later by the successor simulation. Moreover, geostrophic damping, details given in Stipa et al. (2023), is used in the precursor to damp inertial oscillations in the free atmosphere.

A graded mesh in the y- and z-directions is used with  $20 \text{ m} \times 15 \text{ m} \times 10 \text{ m}$  resolution in the wind turbine rotor layer. From the rotor tips, the mesh stretches horizontally to a resolution of 20 m at the lateral boundaries. The mesh is graded vertically; it reduces from uniformly 10 m in the ABL to 5 m in the capping inversion to effectively resolve the trapped waves. The mesh resolution above the capping inversion is uniformly 10 m up to 1 km and stretches above the capping inversion to 200 m resolution at the start of the top of the Rayleigh damping layer. The stretching ratios are 0.9 and 1.1, respectively. Following the recommendations by Khan et al. (2024b), the domain size, advection, and Rayleigh damping layer thicknesses are set greater than one effective horizontal and vertical internal gravity wavelength for the respective directions. The domain size, including the damping layers for all cases, unless mentioned otherwise, is 35 km in the x-direction, 6 km in the y-direction, and 12 km in the z-direction.

The behavior of atmospheric gravity waves can be characterized using non-dimensional flow parameters. Likewise, the configuration of simulations that capture atmospheric gravity waves can also be defined using non-dimensional parameters. The non-dimensional parameters central to this work, as identified by Khan et al. (2024b), are given in Table 1. Here, U is the geostrophic wind speed, N is Brunt-Väisälä frequency, L is length of the wind farm,  $g' = g\Delta\theta/\theta_0$  is a reduced gravitational



**Table 1.** Non-dimensional parameters with ranges used in this study.

|               | Non-dimensional parameter                                                            | Definition              | Range       |
|---------------|--------------------------------------------------------------------------------------|-------------------------|-------------|
| Fr            | Ratio of fluid inertia to buoyant force in the free atmosphere.                      | U/NL                    | 0.075-0.25  |
| $Fr_i$        | Ratio of fluid inertia to buoyant force in the capping inversion.                    | $U/\sqrt{g'H_i}$        | 0.75-1.5    |
| $	ilde{H}_i$  | Ratio of ABL depth to wind farm length.                                              | $H_i/L$                 | 0.025-0.045 |
| $S_h$         | Ratio of rotor tip height to wind farm length.                                       | H/L                     | 0.025-0.048 |
| $\tilde{X}$   | Ratio of computational domain length to horizontal internal wavelength.              | $X/\lambda_{hor}$       | 2.0         |
| $	ilde{L}_z$  | Ratio of computational domain height to vertical internal wavelength.                | $L_z/\lambda_{ver}$     | 1.2         |
| $\tilde{L}_d$ | Ratio of Rayleigh damping layer thickness to vertical internal wavelength.           | $L_d/\lambda_{ver}$     | 1.2         |
| $\tilde{L}_a$ | Ratio of advection damping layer buffer length to horizontal interfacial wavelength. | $L_a/\lambda_{hor-tgw}$ | 1.0         |
| ξ             | Ratio of Rayleigh damping frequency to the free atmosphere buoyancy frequency.       | $1/(\tau N)$            | 1–10        |

constant, g is the usual gravitational constant,  $\Delta\theta$  is the change in potential temperature across the capping inversion,  $\theta_0$  is a reference temperature (usually 300 K),  $H_i$  is capping inversion mid height, H is the wind turbine height (from ground/sea level to top of the rotor), X is the computational domain length,  $L_z$  is non-damped computational domain height,  $L_d$  is the computational domain gravity wave Rayleigh damping layer thickness,  $L_a$  is the computational domain gravity wave advection damping layer buffer thickness,  $\lambda_{hor}$  is the horizontal wavelength of the internal gravity waves,  $\lambda_{hor-ifgw}$  is the horizontal wavelength of the internal gravity waves, and  $\tau$  is the timescale of the gravity wave Rayleigh damping layer. The first four non-dimensional numbers, the Froude numbers for the free atmosphere and capping inversion (Fr and  $Fr_i$ ), and the vertical aspect ratios of the ABL and the wind farm ( $\tilde{H}_i$  and  $\tilde{S}_h$ ), characterize the physics of the flow, whereas the remaining five characterize the simulation configuration.

#### 3.2 Metric for Wave Characteristics and Cr

AGW inclination angles and wavelengths are estimated using a metric that advances an approach proposed by Allaerts et al. (2025) to calculate the inclination of internal waves induced by a hill. The approach of Allaerts et al. (2025) can be applied to the displacement, pressure, and vertical velocity fields; we apply it to only the vertical velocity. A time-averaged vertical velocity plane above the ABL is analyzed to consider only the interfacial and internal waves. The apparent wavelengths of the AGWs can be measured by identifying the local maxima and minima in the vertical velocity field, as shown in Fig. 3 (left) (Khan et al., 2024a). Because wind farms induce two internal wave trains, the upstream one is chosen because it is usually undistorted, especially upstream of the wind farm. Meanwhile, the second wave train superimposes onto the first one, leading to complicated wave characteristics that are difficult to characterize. The maxima/minima finding algorithm identifies even local maxima and minima, so the results are filtered to include only the primary maxima and minima of the first wave train as shown in Fig. 3 (right). While filtering, the first peak, a maximum, is always the one where the wind farm starts and right at the top of the capping inversion, that is, (-3000, 550) m for the case shown in Fig. 3. The minima and maxima locations of the wave

Preprint. Discussion started: 19 November 2025

© Author(s) 2025. CC BY 4.0 License.








field sweep upward and downstream of the first maximum following a roughly parabolic path (Queney, 1948; Allaerts et al., 2025). Thus, a parabola is fit to these locations, starting at the first maximum, always passing through the next minimum, and best fitting the remaining minima and maxima. One AGW metric we wish to obtain is wave field inclination angle. Because the curved parabola has a range of tangent angles, we choose the tangent at the third minimum/maximum from the one at (-3000, 550) m to estimate the representative inclination angle of the first wave train, as shown in Fig. 3 (right). This peak is a maximum corresponding to a vertical location of  $z = H_i + 3L_s/4$ , where  $L_s = 2\pi U/N$  is the buoyancy length. This differs from the observation of Queney (1948) and Allaerts et al. (2025) that the third AGW maximum above the surface occurs at  $z = 5L_s/8$  for their simple ridge flow cases.

The apparent wavelength of internal waves in the direction of energy propagation is calculated using the metric suggested by Khan et al. (2024a). Here, a wavelength is twice the distance between consecutive maxima and minima of vertical velocity along the parabolic fit line. We measure the distance between the consecutive extrema located along the tangent at the third maximum in the  $z = 1/2L_s$  to  $H_i + Ls$  range. This eliminates the possibility of capturing extrema affected by the interfacial waves close to the capping inversion and the propagating internal waves damped by the damping layer beyond one vertical wavelength, which is  $z \ge H_i + L_s$ . The horizontal and vertical wavelengths can be estimated by calculating the streamwise and vertical components of the actual apparent wavelength (i.e. we split the wavelength distance vector into its x- and y-components).

The wavelength of the resultant interfacial wave is calculated using the same metric, but by plotting vertical velocity in the streamwise direction stretching over the length of the wind farm at the top of the capping inversion. The reason for confining the streamwise length in measuring the interfacial wavelength is twofold. First, for low Fr, the two interfacial waves may not or partly superimpose onto each other, modulating the interfacial wave field over the wind farm, and second, the waves downstream of the wind farm are imprints of internal waves for supercritical conditions and a merged form of the imprints and the trapped lee waves for subcritical and critical conditions. Thus, measuring interfacial wavelength over the wind farm length would give a better idea of their involvement in the aerodynamics of the wind farm that induces them.

The amplitude of internal waves is estimated by tracing the path of a Lagrangian particle being carried in the computed velocity field. The path of a particle at  $z = H_i + 3L_s/4$  is traced from the inlet to the outlet. The amplitude is the maximum vertical distance between two consecutive peaks and troughs on the traced particle path. The same strategy is used to estimate the vertical displacement of the capping inversion or the amplitude of the interfacial wave as a function of x. An air parcel is traced along the top of the capping inversion. Thus, local and maximum inversion layer displacements are extracted, which are useful in determining the capping inversion's response to perturbations from individual wind turbine rows and the entire wind farm, respectively.

Accurate simulations must be free of spurious AGW reflections off of domain boundaries. AGW boundary reflection is quantified using the reflection coefficient (Cr), which is estimated with the method used in Khan et al. (2024a) proposed by Allaerts and Meyers (2017). The Cr metric is a modification of the procedure initially given by Taylor and Sarkar (2007). For reflection-free simulations, the criterion Cr 

Figure 3. Time averaged vertical velocity field above the ABL extending up to the top boundary, including the T-RDL beyond  $z/\lambda_{ver} = 1.0$ , where the initially identified peaks are shown with white and filtered peaks with red markers.

#### 3.3 Simulation Suite



The simulation suite given in Table 2 is categorized into three sets, each answering one or two of the research questions outlined in Section 1. The first set compares simulation configurations, truly neutral, rigid lid, and CNBL, used in modeling wind farms under neutral surface conditions. It is established that the CNBL configuration fully resolving AGWs, proposed by Khan et al. (2024b) and adopted for this research, is a more realistic representation of wind-farm-atmosphere interaction. The second set answers how AGW reflections affect the wind farm power output. We show that Cr 



that is,  $L_z=750$  m and 1 km. A further CNBL reference case with the same capping inversion set-up is run, but  $L_z=12$  km to fully resolve AGWs and avoid reflections; and, for conditions defined by Fr=0.15,  $Fr_i=1.3$ ,  $\tilde{H}_i=0.084$ , and  $S_h=0.0255$ .

- (d) NFA-ABL, where the setup is the same as the reference CNBL case, but the free atmosphere is neutrally stratified.
- Set 2 investigates the impact of AGW reflections from the top boundary on the capping inversion displacement and row-averaged wind farm power output. We simulate the wind farm for conditions matching Fr=0.15,  $Fr_i=1.3$ ,  $\tilde{H}_i=0.084$ , and  $S_h=0.0255$ . The non-dimensional damping coefficient of the top Rayleigh damping layer, that is,  $\xi=1/\tau N$ , is varied from 0.1 to 50. The rest of the model configuration is kept the same as described in Section 3.1. In doing so, reflections from the top boundary are controlled and are estimated with the Cr metric. The comparison of capping inversion displacements and power outputs from the simulated cases shows the impact of reflections on flow and power output.
- Set 3 explores three aspects. First, the AGW phenomenon and types (already discussed in Section 2); second, the dependence of AGW characteristics on the physical non-dimensional parameters; and third, the dependence of wind farm performance on the physical non-dimensional parameters. In this regard, several simulations are run for each physical non-dimensional parameter by varying only one at a time and keeping the rest constant. Further details are given in Table 2, Set 3(a-d).

**Table 2.** Details of the simulation sets.

| Set   | Description          | Values of physical non-dimensional parameters |            |              |                | Variables Changed    |                             |  |
|-------|----------------------|-----------------------------------------------|------------|--------------|----------------|----------------------|-----------------------------|--|
|       |                      | $\overline{Fr}$                               | $Fr_i$     | $	ilde{H}_i$ | $S_h$          | Physical             | Simulation                  |  |
| 1 (a) | Truly neutral ABL    | -                                             | -          | -            | 0.0255         | -                    | $L_z$ [12 km]               |  |
| 1 (b) | Rigid Lid            | -                                             | -          | -            | 0.0255         | -                    | $L_z$ [0.5, 0.75 km]        |  |
| 1 (c) | CNBLs                | 0.15                                          | 1.3        | 0.084        | 0.0255         | -                    | $L_z$ [0.5, 0.75, 1, 12 km] |  |
| 1 (d) | NFA-ABL              | -                                             | 1.3        | 0.084        | 0.0255         | -                    | $L_z$ [12 km]               |  |
| 2     | Controlling $Cr$     | 0.15                                          | 1.3        | 0.084        | 0.0255         | -                    | <i>ξ</i> [0.1–50]           |  |
| 3 (a) | Varying $Fr$         | [0.075-0.25]                                  | 1.3        | 0.084        | 0.0255         | N                    | -                           |  |
| 3 (b) | Varying $Fr_i$       | 0.15                                          | [0.25–1.5] | 0.084        | 0.0255         | $\Delta \theta$      | -                           |  |
| 3 (c) | Varying $	ilde{H}_i$ | 0.15                                          | 1.3        | [0.043-0.26] | 0.0255         | $H_i, \Delta \theta$ | -                           |  |
| 3 (d) | Varying $S_h$        | 0.15                                          | 1.3        | 0.084        | [0.0255-0.048] | H                    | -                           |  |

Preprint. Discussion started: 19 November 2025

© Author(s) 2025. CC BY 4.0 License.

#### 285 4 Results



This section analyzes three major aspects of modeling the interaction of wind farms with the atmosphere. First, three common approaches to configuring wind farm simulations for neutral ABL conditions are compared to distinguish the most realistic one. As will be shown later in this section, accurate wind farm performance predictions require simulation configurations that include a significant part of the stratified free atmosphere above the capping inversion, thus fully resolving AGWs. Second, the impact of AGW reflections on wind farm flow is investigated to ensure the accuracy of the CNBL setup. Third, the dependence of AGW characteristics and wind farm performance on the physical non-dimensional parameters is analyzed to better understand the underlying phenomena of wind farm interaction with the atmosphere.

#### 4.1 Comparison of Wind Farm Simulation Configurations

Three commonly used wind farm flow simulation configurations, truly neutral, rigid-lid, and CNBL, are compared in this section. The aim in this regard is to identify any shortcomings associated with each configuration in accurately mimicking the wind-farm-atmosphere interaction. The analysis to compare these configurations is done on the simulations detailed in Table 2, Set 1 (a-d). CNBL conditions are defined by Fr = 0.15,  $Fr_i = 1.3$ ,  $\tilde{H}_i = 0.084$ , and  $S_h = 0.025$ , which are irrelevant for the other two configurations.

#### 4.1.1 Truly Neutral and Rigid Lid Configurations

300 Truly neutral configurations are helpful when the interaction of the wind farm with only a neutral ABL is of interest. This approach is somewhat of an academic curiosity and might not occur realistically, as it assumes that the atmosphere, even beyond the turbulent ABL, is neutral. It can be of use to study turbines with heights much smaller than the ABL. However, modern wind turbines in large wind farms can interact with the atmosphere above the ABL (Allaerts et al., 2018). Still, many LES studies have used the TNBL configuration, where a pressure gradient or constant geostrophic wind drives the boundary layer. (Wu and Porté-Agel, 2015; Stieren and Stevens, 2022)

For a TNBL simulation, a constant potential temperature in a domain a few times taller than the intended ABL height is simulated. The ABL can freely expand vertically until an equilibrium with the non-turbulent free atmosphere is achieved. Therefore, the truly neutral simulation configuration detailed in Table 2, Set 1 (a) has an ABL height of  $H_i = 500$  m at the inflow boundary, and the domain height is  $L_z = 12$  km. The domain height matches that of the reference case (i.e. CNBL configuration fully resolving AGWS), but the simulation does not use a top RDL because there is no buoyancy to trigger AGWs in the truly neutral case. As shown on the vertical velocity plots in Fig. 4, the ABL displaces vertically in reaction to horizontal flow deceleration through the wind farm without being constrained by the rigid top boundary. This is also confirmed in Fig. 5, as the particle path traced through the domain starting at an initial ABL height of 500 m at the inflow initially rises and then asymptotes at a height far lower than the domain height. Moreover, this particle path shows the highest vertical expansion among the cases investigated, as there is no capping inversion to cap the vertical displacement of the ABL.

Figure 4. Temporally averaged vertical velocity through and around the wind farm with lengths L=5.67 km from simulation configurations characterizing rigid lids, CNBLs, truly neutral, and the idealized configuration with both the ABL and free atmosphere being neutral. The horizontal dotted line shows the capping inversion mid-height where implemented.

Like truly neutral cases, the rigid-lid configurations have a constant potential temperature, but the domain height is defined at the inversion layer or ABL height. In a physical sense, the top boundary acts like an immovable capping inversion that constrains the vertical displacement of the flow in reaction to the horizontal flow deceleration caused by the wind farm. This constraint on vertical velocity suggests that, to enforce continuity, the flow cannot decelerate as it naturally would. The closer the top boundary is placed to the top of the turbines, the greater this effect on the wind farm. The fact that the top boundary placement decreased horizontal flow deceleration through the wind farm can be witnessed on the vertical velocity contours




Figure 5. Particle vertical displacement,  $\eta$ , versus horizontal distance along the domain at capping inversion mid-height  $L_z=500$  m, thus representing the displacement of an air parcel perturbed by the wind farm and analogous to capping inversion displacement for the cases simulating one. The wind farm stretches from x/L=-0.5 to 0.47.

for the rigid lid simulations shown in Fig. 4, where the domain heights are 500 m and 750 m. The particle paths for the rigid lid cases shown in Fig. 5 are concrete evidence of an immovable top boundary; vertical flow is completely suppressed when  $L_z=500$  m compared to the truly neutral case. This "channel" effect is milder for the rigid-lid  $L_z=750$  m case but is very pronounced compared to the truly neutral and the reference cases that use much taller domains.

# 4.1.2 Partial-to-Fully Resolved CNBL Configurations

The discussion in Section 4.1.1 clearly shows the importance of simulating domains tall enough to exclude channel effects caused by too low a top boundary. However, temperature conditions beyond a few hundred meters above the sea surface are not always neutral; instead, a stable capping inversion and free atmosphere aloft are frequent. Thus, the discussion naturally leads to whether or not an inversion layer and the free atmosphere should be included in wind farm simulations. Moreover, to what extent should the free atmosphere be resolved to capture all phenomena relevant to wind-farm–atmosphere interaction? These questions can be answered by comparing CNBL configurations that partially resolve AGWs with a configuration that fully resolves them; these configurations are detailed in Table 2, Set 1 (c). Only inlet Rayleigh and advection damping layers are implemented for CNBL-750 m and CNBL-1000 m setups, as there is barely any free atmosphere resolved for the internal waves to form.

Besides properly resolving AGWs, avoiding the effect of the top boundary to constrain the capping inversion displacement is critical. The evidence of unresolved, partially, and fully resolved AGWs can be seen on the temporally averaged vertical velocity contours for three CNBL cases,  $L_z=750$  m, 1000 m, and 12 km, shown in Fig. 4. No AGWs are seen for the case with  $L_z=750$  m, traces of interfacial waves are seen when  $L_z=1000$  m, and all AGWs are resolved when  $L_z=12$  km. Since AGWs perturb the capping inversion, improperly resolved AGWs will show unrealistic impacts on the flow in the ABL. In less tall computational domains, improperly resolved AGWs and the channel effect both contribute to unrealistic flows in the ABL. This can be appreciated from the particle paths for CNBL cases shown in Fig. 5. A particle traced at the top of the capping inversion shows suppressed vertical movement for the cases with less tall domains compared to the tallest one. Placing the top

Preprint. Discussion started: 19 November 2025

© Author(s) 2025. CC BY 4.0 License.






boundary close to the capping inversion accelerates the flow in the free atmosphere and suppresses the realistic displacement of the capping inversion.

Fully resolving AGWs requires a computational domain much taller (i.e.  $L_z=2L_s$ ) than one sufficient to eliminate the constraining channel effect of the top boundary. This is obvious when the particle path for the case of fully resolved AGWs is compared to the truly neutral case. The particle path for the case of fully resolving AGWs characterizes the perturbations from the wind farm and the AGWs. The ABL does not expand progressively, and flow acceleration effects from the top boundary are absent. The global maxima and minima on this path are evidence of two disturbance sources, each inducing a wave train in the free atmosphere. The fact that the path is wavy downstream of the wind farm shows the resultant interfacial wave, which is the result of superimposed internal waves in that region and the interfacial waves advected downstream for this supercritical case. Therefore, the configuration fully resolving AGWs is most realistic and accurate in mimicking wind-farm-atmosphere interaction for shallow ABL conditions (i.e. CNBLs).

# 4.1.3 Capping Inversion Between Neutral Free Atmosphere and ABL

The discussion on CNBL configurations in Section 4.1.2 asks whether modeling a stratified free atmosphere is essential. This curiosity is settled by simulating a configuration similar to the case that fully resolves the AGWs, with the key difference that free atmosphere potential temperature profile is neutral, details of which are given in Table 2, Set 1 (d). The only stable layer is the capping inversion. Figure 4 shows that there are no internal waves, but trapped lee waves at the capping inversion are present. We use no damping layers in this simulation setup. The significance of including the free atmosphere's stability can be appreciated from the particle path traced for this scenario. The path is similar to that of the case that fully resolves AGWs up to the second row of the wind farm; beyond that, the capping inversion slowly moves upwards and approaches that of the truly neutral case far downstream. This rising tendency of the capping inversion in response to the wind farm perturbation is because the buoyancy of the free atmosphere is absent to mask it. In the physical sense, the free atmosphere's buoyancy acts like a spring's stiffness; the buoyancy of the free atmosphere restores the displaced capping inversion. Without buoyancy, the restoring force that suppresses capping inversion vertical displacement is missing. The impact of flow entrainment at the end of the wind farm on the capping inversion is almost unnoticeable. Moreover, the wavy path downstream of the wind farm is the signature of trapped lee waves only because there are no internal waves.

## 4.1.4 Comparison of Wind Farm Power for the Configurations

Figure 6 (a) compares wind farm power output for the configurations described earlier. The solid lines show the averaged power output of the wind turbines, where we average both in time and along spanwise turbine rows. The dashed lines show the difference between averaged power output for each configuration and that of the case that fully resolves AGWs (the CNBL-12 km case) assuming it is the most realistic. Several conclusions can be drawn from these power plots and their comparisons. First, the power output follows the same trend along the wind farm length, regardless of the configuration used. A few notable aspects of this trend include the greatest power decrease for the turbines in the second row, increasing power output till the fifth row, and slightly decreasing towards the last row. The highest drop for the second turbine is because of the velocity



deficit caused by the first turbine, and little to no turbulent flow entrainment from above. The increasing power output after the second row suggests better wake mixing and recovery from the sides and above the wakes. The flow has fully developed by the fifth row and mixing has reached equilibrium. The truly neutral simulation gives the best perspective of the power for turbines beyond the fifth row, as it excludes flow "channeling" effects caused either erroneously by too low a top boundary or realistically by inversion layer displacement effects.

Comparing the configurations, we observe that the rigid-lid cases appear to have the most overestimated wind farm power output, inferred in the above discussions as the consequence of the channel effect caused by too low a top boundary. It is further inferred that the power curves for the rigid-lid cases would approach that of the truly neutral case if one keeps increasing the domain height. Surprisingly, the power profile predicted by the CNBL cases, partially resolving the AGWs, is almost the same as that of the rigid-lid case with the  $L_z=750$  m. This shows that including a capping inversion without sufficiently resolving the free atmosphere has the channeling effect almost completely dominating the stratification effects. The power plot for the case where the free atmosphere was kept neutral above the capping inversion of the same strength as that of the CNBL cases is interesting: this plot roughly matches the reference tall CNBL case until the fourth row. Beyond the fourth row, the simulated power is less than the reference case, signifying the absence of favorable pressure gradients in that region caused by internal waves and the capping inversion constraining the vertical flow mixing in the ABL when compared to the truly neutral case. This mismatch shows the importance of modeling a stable, free atmosphere.

Figure 6. Temporal and row-averaged power as a function of (a): domain height for fully neutral, Rigid lid, CNBL, and NFA-ABL configurations to mimic neutral surface conditions, and (b):  $\xi$ . The solid lines show actual power, whereas the dashed lines show the difference between the actual power predicted by each configuration and that of the reference case for (a) and  $\xi = 10$  for (b).

The dashed lines show that all configurations, except the one with a neutral free atmosphere, underestimate the global blockage effect and overestimate wind farm power output compared to the reference case that fully resolves AGWs. The higher power produced by the first row turbines in all cases compared to the reference case is evidence of the underestimated

Preprint. Discussion started: 19 November 2025

© Author(s) 2025. CC BY 4.0 License.




global blockage effect. Interestingly, the simulated power of the reference case approaches that of the truly neutral case towards the far end of the wind farm.

#### 4.2 Impact of Reflections on Wind Farm Flow

AGWs will reflect off of domain boundaries if not properly treated. Reflected waves then affect the flow in the free atmosphere, in turn affecting the displacement of the capping inversion and the flow in the ABL. Hence, the non-physical reflections from the domain boundaries can affect the accuracy of the flow solution. AGWs occur mainly in the free atmosphere, and a major source of spurious reflection is the top boundary. Spurious reflection can occur at the inlet, also, but we use an advection-damping layer there, as explored by Khan et al. (2024b), with an optimal setup. The impact of the reflections from the top boundary on the capping inversion displacement and wind farm power output requires more attention. The reflections from the top boundary are controlled by setting the time scale,  $\tau$ , of the Rayleigh damping layer at the top boundary (T-RDL). As shown on the temporally averaged vertical velocity plots in Fig. 7, the AGWs reflect the most from the top boundary when the damping strength is the weakest. Damping strength is described by the non-dimensional  $\xi = 1/(\tau N)$ . For the range of  $\xi$  from 1.0–10.0, the waves are damped significantly, and the AGW amplitude decreases towards the top boundary. Using  $\xi > 10$  makes the T-RDL too strong and the AGWs reflect off it, as seen in Fig. 7 for  $\xi = 50$ .

The particle paths shown in Fig. 8 give insight into the impact of reflections on the capping inversion displacement. For  $\xi = 1.0$  to 10, the capping inversion displacements differ only slightly because the reflections are well-controlled, i.e., Cr = 8-9%. In contrast, the capping inversion displacement is most different for  $\xi = 0.1$ , especially towards the end of the wind farm and downstream. This is because the wave reflection angle depends on the wave propagation inclination angle, and for the simulated conditions, the reflected waves fall more downstream of the wind farm. For  $\xi = 50$ , too large a damping strength, the waves reflect off the T-RDL. This can be seen in the capping inversion displacement that looks different all across the domain length than in the case in which reflection is well controlled. This means the waves are reflected from the T-RDL across the length of the domain, and the reflected waves contribute to perturbing the capping inversion, which is not desired.

The impact of reflections from the T-RDL on the wind turbine and farm output is rather mild and relates to the capping inversion displacement. As shown in Fig. 6 (b), the power plots are the same in all cases except for underestimated power towards the end of the wind farm for  $\xi = 0.1$ . The temporal and row-averaged power looks slightly different from other cases throughout the wind farm length for  $\xi = 50$ , where the power is overestimated for the first turbine and underestimated for all other turbines except the fifth one.

Figure 7. Contours of temporally averaged vertical velocity in a streamwise-oriented vertical plane for varying Rayleigh damping strength ( $\xi$ ) of the T-RDL. The flow conditions are defined by  $Fr=0.15, Fr_i=1.3, \tilde{H}_i=0.087,$  and  $S_h=0.0255.$ 

Figure 8. Capping inversion displacement, extracted by tracing particle paths at z = 550 m, as a function of  $\xi$ .

Preprint. Discussion started: 19 November 2025

© Author(s) 2025. CC BY 4.0 License.





# 4.3 Non-Dimensional Parameter Dependence of Capping Inversion Layer Shape and AGW Characteristics

With optimal T-RDL strength established, we now address the dependence of the AGW characteristics on the physical non-dimensional parameters listed in Table 1. The following subsections discuss capping inversion shape and interfacial and internal wave characteristics as a function of the physical non-dimensional parameters. The capping inversion shape at steady state shows how the flow in the ABL, capping inversion, and free atmosphere are coupled Stipa et al. (2024). We estimate the capping inversion shape from temporally averaged streamwise and vertical velocity fields. We examine AGW characteristics using the vertical velocity field. The methods used to measure these representative features should always be used with a thorough visual analysis of flow fields. Note that the domain width was set to 18 km for the case  $\tilde{H}_i = 0.042$ . The initially used 6 km wide domain was too narrow for simulating the capping inversion at a height of only 250 m. The AGWs are strong in shallow boundary layers and high wind veer recycled the AGWs from the lateral boundaries into the wind farm, effecting its power output.

# 4.3.1 Capping Inversion Displacement Dependence on Physical Non-Dimensional Parameters

AGW characteristics are strongly related to the displacement of the capping inversion, which can be depicted by its shape at steady state. For instance, the positive and negative displacements of the capping inversion at the start and the end of the wind farm are a result of the two main flow disturbances that trigger gravity waves, the bulk upflow and downflow at the upstream and downstream wind farm edges, respectively. Moreover, the wavy capping inversion shape downstream of the wind farm shows the resultant interfacial waves. Each positive and negative vertical displacement of the capping inversion is inherently linked to diverging and converging flows underneath in the ABL. The size and strength of these convergent and divergent ABL flow zones are related to AGW characteristics. With a convergence comes a horizontal flow acceleration; with a divergence comes a deceleration.

After initial temporal transients, the capping inversion settles into an equilibrium shape that depends on the flow conditions and wind farm size, which we describe with the physical non-dimensional parameters. The capping inversion height varying in streamwise direction as a function of the physical non-dimensional numbers, is shown in Fig. 9. In analyzing these graphs, the focus is on the shape of the capping inversion at various streamwise locations to relate it to the global blockage effect and favorable and unfavorable pressure distributions through and downstream of the wind farm. As described in Section 4.1.2, the capping inversion shape is computed by tracing a particle path along the capping inversion top. As seen in Fig. 9, the general capping inversion shape has global maxima, always at around the wind farm start, linked to the horizontal flow deceleration caused by the wind farm. Conversely, global minima are usually at the end of the wind farm, thus linked to the acceleration of the horizontal flow in the exit and wake regions of the wind farm. In some cases, wave interference leads to complex capping inversion shapes. The maximum capping inversion displacement is also measured from its shape by differencing the highest and lowest vertical positions on the particle path. The variations in interfacial wavelengths for varying values of the non-dimensional parameters can also be appreciated from these capping inversion shapes.

Figure 9. Capping inversion displacement along the streamwise direction as a function of Fr,  $Fr_i$ ,  $\tilde{H}_i$ , and  $S_h$ . Unlike the rest, data points  $Fr_i = 0.25$  and 0.5, given for completeness, are created from simulations with laminar inflows.

**Figure 10.** Maximum inversion layer displacement as a function of Fr,  $Fr_i$ ,  $\tilde{H}_i$ , and  $S_h$ .

The maximum capping inversion displacement normalized by turbine diameter  $(CI_{disp}/D)$  as a function of Fr,  $Fr_i$ ,  $H_i$ , and  $S_h$  is shown Fig. 10.  $CI_{disp}/D$  increases abruptly for increasing Fr until 0.0125, increases only slightly for larger Fr. The physical meaning of increasing maximum displacement can be inferred as reduced atmospheric stability or milder response by buoyancy. A disturbance can displace the capping inversion slightly more than it could for a more stable-free atmosphere. The trend is not monotonic around Fr = 0.15, though, which is likely caused by the complex interference of the two internal wave trains.  $CI_{disp}/D$  is maximum for  $Fr_i = 1.0$ , as positive capping inversion displacement is the highest for critical conditions. Note that only one value of  $Fr_i < 1.0$  is practical because values of  $Fr_i < 0.75$  require impractical capping inversion strengths when  $H_i$  is kept constant. The estimated capping inversion displacements from the two simulations with impractical capping inversion strengths (i.e.,  $Fr_i = 0.25$  and 0.5) confirm capping inversion displacement smaller than that of the  $Fr_i = 1.0$  case. In a physical sense,  $Fr_i$  approaching 0 would mimic a rigid lid that would barely move due to the disturbance from a wind farm.

On the other hand, the decreasing  $CI_{disp}/D$  for values of  $Fr_i > 1.0$  is because of background flow washing away buoyancy effects, thus weakly perturbing the capping inversion.

The maximum capping inversion displacement as a function of the ABL-aspect ratio shows an intuitive behavior. The maximum capping inversion displacement decreases for increasing  $\tilde{H}_i$  values because the capping inversion is further from the wind farm. Thus, the global minimum of capping inversion displacement in the wind farm wake region is barely noticed for values of  $\tilde{H}_i > 0.17$ . The maximum capping inversion displacement is mildly sensitive to  $S_h$  as placing the wind turbines closer to the capping inversion perturbs it slightly more than the rotors at standard hub heights, i.e. 90 - 150m, but still negligible

495

500

when normalized with the rotor diameter. Note that this observation is in the context of the wind farm layout and turbine size simulated in this study. We infer that a wind farm of higher power density and one with bigger turbines might be stronger momentum sinks, causing bigger capping inversion displacements.

# 475 4.3.2 Non-Dimensional Parameter Dependence of Interfacial Waves

In Section 4.3.1, we indicate that the properties of AGWs depend on the physical non-dimensional parameters. This section analyzes the dependence of interfacial wave characteristics on these parameters.

Interfacial waves are bound to the capping inversion, and their wavelengths are correlated to the sizes of converging and diverging zones in the ABL. Thus, the average interfacial wavelengths can be measured along the inversion layer top by using the following expression.

$$\lambda_{ifgw} = \frac{2\sqrt{(x_e - x_s)^2 + (y_e - y_s)^2}}{n_{peaks}},\tag{1}$$

where  $(x_s, y_s)$  and  $(x_e, y_e)$  are the coordinates of the first and last peaks along the capping inversion that extends from the wind farm entrance to exit, respectively, and  $n_{peaks}$  is the total number of peaks (i.e. maxima and minima). Since it is measured in the streamwise direction, it is the average horizontal wavelength. We measure the interfacial wavelength over the length of the wind farm, as it is more suitable for this study.

The interfacial wavelengths normalized with wind farm length  $(\lambda_{ifgw}/L)$  as a function of Fr,  $Fr_i$ ,  $\tilde{H}_i$ , and  $S_h$  are shown in Fig. 11. The interfacial wavelengths based on the deep-water approximation are also calculated and presented for comparison. This approximation excludes the impact of ABL depth on the interfacial wavelengths and is valid when  $kH_i \gg 1$ , where k is the horizontal wavenumber given by Sachsperger et al. (2015) as

$$k = \frac{g'}{2U^2} + \frac{N^2}{2g'}$$
. (2)

The horizontal interfacial wavelength calculated from the deep-water equations is then  $\lambda_{ifgw}=2\pi/k$ , which depends on the reference potential temperature and the temperature jump across the capping inversion, and the stability in the free atmosphere. Clearly, the calculated  $\lambda_{ifgw}$  is independent of the boundary layer height, thus applicable to deep boundary layers only.

As can be visually seen in Fig. 12, the measured  $\lambda_{ifgw}$  increases for increasing Fr; the interfacial waves become skewed, which means the height of the darkest-colored regions at the bottom of the contours, which is representative of the interfacial waves, shrinks. Even the propagating internal waves are strongly constrained from propagating vertically, so the wave field appears tilted more downstream. The measured interfacial wavelengths are observed to be half the wind farm length for the lowest Fr and gradually increase to about full wind farm length for the highest Fr. Thus, the sizes of zones of favorable and unfavorable pressure gradient in the ABL vary between 0.25L and 0.5L for increasing Fr. The calculated  $\lambda_{ifgw}$  for Fr also shows an increasing trend but is distinct from that of the measured one. The calculated values agree with the measured only for


Figure 11. Interfacial wavelengths normalized with wind farm length as a function of Fr,  $Fr_i$ ,  $\tilde{H}_i$ , and  $S_h$ . A comparison of wavelengths measured from LES data with those calculated with the deep-water approximation, where the ABL depth is considered large enough and therefore insignificant.

low values of Fr. This could be because  $kH_i$  for constant  $H_i = 500 \ m$  is significantly greater than unity for only low values of Fr.

Both measured and calculated  $\lambda_{ifgw}$  increase roughly linearly with  $Fr_i$  until an asymptote at 0.75~L for the measured and at L for the calculated is reached when  $Fr_i=1.25$ . There is good agreement between measured and calculated for sub-critical conditions, but a discrepancy arises beyond critical conditions ( $Fr_i \geq 1.0$ ). The matching values of  $\lambda_{ifgw}$  are mainly because it varies in response to only varying  $\Delta\theta$  in setting  $Fr_i$  while holding  $H_i$  and N fixed at 500~m and  $0.01~s^{-1}$ , respectively. We recall from Section 4.3.1 that the capping inversion becomes more sensitive for sub-critical values of  $Fr_i$  and interfacial waves are smaller than for super-critical values. For instance, the two laminar inflow cases mentioned in Section 4.3.1 (i.e.  $Fr_i=0.25~m$  and 0.5) showed that  $\lambda_{ifgw}$  approaches the streamwise turbine distancing as if each turbine row is triggering an interfacial wave. Whereas, for super-critical conditions, the capping inversion perturbs on length scales increasing between half to three-quarters of the wind farm length. This means that the sizes of zones of favorable and unfavorable pressure gradient in the ABL vary between row-distancing and 0.4L for increasing  $Fr_i$ .

Figure 12. Temporally averaged vertical velocity plots, inclination angles with the horizontal axis, and peaks at maximum and minimum vertical velocities fitted with a parabolic curve. The vertical planes in a column are for the varying value of Fr,  $Fr_i$ ,  $\tilde{H}_i$ , or  $S_h$ , while the other parameters were kept constant. Note that plots for  $Fr_i = 0.25$  and 0.5 are taken from simulations driven by uniform inflows, given for completeness.

Preprint. Discussion started: 19 November 2025

© Author(s) 2025. CC BY 4.0 License.





Unlike the calculated  $\lambda_{ifgw}$ , the measured values show an increasing trend against  $\tilde{H}_i$  until they reaches a maximum of about  $\lambda_{ifgw}/L=1.0$  for  $\tilde{H}_i=0.17$ . The interfacial wavelengths become slightly shorter for further increasing  $\tilde{H}_i$ . This shows that the wind farm is seen as a single disturbance beyond a certain ABL-aspect ratio, and the capping inversion receives a weaker and spread-out disturbance. The calculated  $\lambda_{ifgw}$  shows the increasing trend only until  $\tilde{H}_i=0.084$ , beyond which it shows a linearly decreasing trend with an almost perfect match between calculated and measured for  $\tilde{H}_i=0.125$ . These observations indicate that the deep-water approximation follows these simulations only for deep ABLs. The sizes of zones of favorable and unfavorable pressure gradient in the ABL vary between 0.25L and 0.4L for increasing  $\tilde{H}_i$ . As expected,  $S_h$  has a negligible impact on the interfacial wavelengths; both measured and calculated  $\lambda_{ifgw}$  are independent of it. The  $\lambda_{ifgw}$  is the same as that of Fr=0.15.

Generally, the amplitude of interfacial waves varies in the narrow range of 0.19D-0.3D, where D is rotor diameter, against the physical non-dimensional parameters, except for  $\tilde{H}_i$ , where the range is 0.02D-0.6D, which is shown in Fig. 10. For low values of  $\tilde{H}_i$ , the capping inversion displacement is higher, and strong waves are present, as the two are directly proportional (Allaerts et al., 2018). Amplitudes of interfacial waves are analogous to capping inversion displacements, which are already reported in Section 4.3.1. Moreover, the inclination angles of interfacial waves with respect to lateral axis are dependent only on  $Fr_i$  and have already been investigated by Lanzilao and Meyers (2024).

# 4.3.3 Non-Dimensional Parameter Dependence of Internal Waves

The propagating internal waves are relatively weaker in strength than the interfacial waves, and their properties depend mainly on Fr and  $Fr_i$ . A major challenge in measuring their properties is that there are two internal wave trains superimposing on each other, one from the front of the farm and one from the rear. Especially for high Fr, the superimposition is evident as the inclination angle is expected to be higher. Therefore, we focus on the first wave train, as we did in Section 3.2, as it is less distorted.

Figure 13 shows wave field inclination angle, as defined in Section 3.2, extracted from average vertical velocity fields as a function of the four physical non-dimensional parameters. The inclination angle shows a decreasing trend against increasing Fr starting at around  $63^{\circ}$ , for Fr = 0.075, and approaching around  $27^{\circ}$  for Fr = 0.25. The reduced slope for  $Fr \geq 0.15$  of the line connecting the data points means that the two internal wave trains are more distinct; for higher values of Fr, the first wave train stretches across the length of the domain, see in Fig. 12. The inclination angle has the same decreasing trend with increasing  $Fr_i$ , decreasing from  $61^{\circ}$  to  $40^{\circ}$ . This  $40^{\circ}$  value is roughly the same as for Fr = 0.15, and in the varying  $Fr_i$  cases, Fr is held at 0.15. The inclination angle of the internal waves seems almost independent of  $\tilde{H}_i$  and  $S_h$ . There is a variation for the former, but in a narrow range of  $41 - 35^{\circ}$ , probably caused by varying capping inversion displacement against  $\tilde{H}_i$ .

The apparent wavelength, measured along the fit parabola, of the first wave train is dependent on its inclination angle. In measuring this wavelength, it is assumed that the interference of the two wave trains is minimal for values of  $Fr \leq 0.15$ . We also recall that the measured wavelength with the method proposed in Section 3.2 allows us to estimate horizontal and vertical components of the actual wavelength. Thus, the effective horizontal wavelength could be  $\lambda_a = 2\lambda_a \cos \phi$ , equivalent to the sum of horizontal wavelengths of the two dominant wave trains. The apparent wavelength of the first wave train, normalized with L

**Figure 13.** Inclination angle of internal waves as a function of Fr,  $Fr_i$ ,  $\tilde{H}_i$ , and  $S_h$ .

**Figure 14.** Actual wavelength of internal gravity waves as a function of Fr,  $Fr_i$ ,  $\tilde{H}_i$ , and  $S_h$ .



and as a function of the physical non-dimensional parameters, is given in Fig. 14. Generally, this wavelength increases linearly and is smaller than the wind farm length for values of  $Fr \leq 0.15$ . This linear increase is mainly due to increasing buoyancy length (i.e.  $L_s$ ), which is directly proportional to Fr. Note that the first wave train is dilated and merged with the second for Fr > 0.15 by the increasing background speed, and only one wave train is obvious in the solution, as shown in Fig. 12.

As stated at the start of this section, the internal wave properties are mainly affected by Fr. This is obvious when the apparent wavelength plots for the remaining physical non-dimensional parameters are analyzed, where Fr=0.15 while varying each. Thus,  $\lambda_{igw}/L$  varies in the narrow range of 0.9L-1.1L for the simulated values of  $Fr_i$ ,  $\tilde{H}_i$ , and  $S_h$ . The values reported for  $Fr_i=0.25$  and 0.5 are approximated, as negligible internal waves were observed for these idealized cases, as can be seen in Fig. 12. Small variations about the average value (i.e. 1.0L), which is roughly the value for Fr=0.15, for varying  $Fr_i$ ,  $\tilde{H}_i$ , and  $S_h$  are the impacts of the interference of the two wave trains that depend on the inclination angles. For instance, the  $\lambda_{igw}/L$  plot against  $Fr_i$  resembles the decreasing inclination angle trend against  $Fr_i$  shown in Fig. 13. In the case of  $\tilde{H}_i$ , the wavelength only mildly increases with increasing  $\tilde{H}_i$ , where for the shallowest boundary layer simulated, the wavelength is slightly larger than that for Fr=0.15. For low  $\tilde{H}_i$ , the capping inversion feels both disturbance sources more isolated and strongly, but beyond  $\tilde{H}_i=0.17$ , the wind farm is an entity, as the growing internal boundary layer rather than the two disturbance sources perturb the capping inversion. Thus, the wavelength resembles a dilated single disturbance. There is barely any change in the wavelength against  $S_h$ , as all simulated  $S_h$  values yield a wavelength as wide as 1.0L, a virtue of the constant Fr=0.15.

Figure 15. The amplitudes of internal waves measured as the amplitude of streamline displacement at a height  $H_i + 3/4L_s$  as a function of Fr,  $Fr_i$ ,  $\tilde{H}_i$ , and  $S_h$ .

https://doi.org/10.5194/wes-2025-236 Preprint. Discussion started: 19 November 2025






The amplitudes of internal waves, which are measured as the amplitude of streamline displacement, as a function of the physical non-dimensional parameters, are shown in Fig. 15. The amplitudes are measured by tracing a particle at  $z=H_i+3/4L_s$ . There is little variation in the amplitudes as a function of Fr, except for a small increase from 0.11D to 0.14D between Fr=0.075 and 0.125. The amplitude increases as a function of  $Fr_i$  only until 0.17D, suggesting that the capping inversion becomes less rigid until the amplitude depends only on Fr. There is an almost linearly decreasing amplitude trend against  $\tilde{H}_i$ , going from 0.19D for  $\tilde{H}_i=0.025$  to 0.08D for  $\tilde{H}_i=0.25$ . The reason for weaker internal waves for increasing  $\tilde{H}_i$  was discussed in Section 4.3.1. The amplitudes only slightly increase over 0.17D against  $S_h$ , which is the value of amplitude for Fr=0.15.

## 4.4 Non-Dimensional Parameters Dependence of Wind Farm Aerodynamics

Here, we analyze and discuss the dependence of wind farm performance on the physical non-dimensional parameters that define CNBL flow conditions. The capping inversion displacement couples the flow in the ABL with the capping inversion and the free atmosphere aloft. In this section, the simulated wind turbine power as a function of distance downstream is investigated and related to the capping inversion displacement. It is hypothesized that the local capping inversion displacement affects the power output of the wind turbines below, mainly as an additional effect to the prevailing induction effect for the first row and wake effects for the remaining rows. Thus, an element of the global blockage effect is a consequence of the capping inversion displacement upstream to the first row of the wind turbines as a result of the wind farm's solidity and energy extraction from the atmosphere. Moreover, the internal and interfacial waves, the latter only under sub-critical conditions, may contribute to the global blockage effect, as they can propagate upstream. This is strictly in the case that upstream propagation of these waves affects the capping inversion displacement, which is difficult to measure in an isolated manner. However, it is obvious that the first interfacial wave strongly contributes to the blockage effect. Often, it is stretched downstream, which affects the localized power output of the first few turbine rows. For values of Fr > 0.15, internal waves tilt downstream significantly and may superimpose onto the interfacial waves, affecting localized capping inversion displacement and impacting the localized power output. The enhanced wind farm wake recovery is mainly associated with the dropping of the capping inversion at the end of the wind farm. Under supercritical conditions, internal waves leave an imprint in the velocity field of the wake of the wind farm. Trapped lee waves dominate this imprint for  $Fr_i \leq 1$ . Adverse and favorable pressure gradients are correlated with the gradient of the capping inversion displacement, which varies in the streamwise direction. The dominant pressure variations are due to the capping inversion displacement caused by upward flow deflection at the entrance of a wind farm and downward deflection in its wake. The intensity of the pressure gradients is most sensitive to the ABL depth. For high values of ABL-aspect ratio, these gradients are relatively mild. In the following subsections, all of these preliminary claims will be supported by simulated turbine power outputs.

### 4.4.1 Wind Farm Power Output as a Function of Physical Non-Dimensional Parameters

Here, we discuss how the global blockage effect and pressure variations relate to each non-dimensional parameter via the capping inversion displacement height at steady state. Generally, the localized capping inversion displacement has a negative


correlation with the row-averaged power output. This means that a higher capping inversion corresponds to decreased power of the row underneath it and vice versa. Temporal and row-averaged power output are analyzed to uncover general trends in wind farm performance in the maximum wake loss scenario (i.e., when wind is straight down the rows). Figure 16 (a) shows the temporal and row-averaged wind turbine power output as a function of Fr and streamwise distance. Wind farm blockage decreases as Fr is increased. However, a narrow variation of  $0.2 \ MW$  is observed for the power output of the 1st row over the simulated values of Fr. It is astonishing how the power output of the second row is completely independent of Fr as the wake effects dominate. Beyond the second row, the row-wise power output varies about a trend characteristic of aligned wind farm aerodynamics, with a maximum at the fifth or sixth row. Although subtle, variations are bound to the localized capping layer displacement aloft as discussed in Section 4.3.1. Generally, row-wise power output decreases beyond the fifth row until a slight increase is observed for the last two rows. The power output of individual rows beyond the second decreases slightly with increasing Fr except for Fr = 0.075 due to its unique capping inversion shape, which is shown in Fig. 9.

Figure 16. Temporal and row-averaged wind turbine power output as a function of Fr,  $Fr_i$ ,  $\tilde{H}_i$ , and  $S_h$ .

Figure 16 (b) shows the row-wise power output for a few values of  $Fr_i$ . The wind farm blockage effect reduces with increasing  $Fr_i$ , where the strongest blockage is observed for the subcritical case (i.e.  $Fr_i = 0.75$ ), but is almost the same as the critical case. The simulated power of the second row is the same for values of  $Fr_i \ge 1.0$  but higher for  $Fr_i = 0.75$  because

https://doi.org/10.5194/wes-2025-236 Preprint. Discussion started: 19 November 2025

© Author(s) 2025. CC BY 4.0 License.





the capping inversion displacement is negative in contrast to positive in the remaining cases. The fourth row produces the highest power for  $Fr_i = 0.75$  among the waked rows because of the localized favorable pressure, as the capping inversion in the vicinity is at its lowest point. The highest power output among the waked rows is from the fifth-row turbines for the critical and supercritical cases. The simulated power shows a gradual decrease past the fifth row for values of  $Fr_i \ge 1.0$  till the seventh row, beyond which it increases slightly. The fact that power oscillates with distance downstream for  $Fr_i = 0.75$  is evidence of power's negative correlation to the localized capping inversion displacement. This also confirms that gravity waves can cause several zones of converging and diverging flow through the wind farm.

The simulated row-wise power profile is the most sensitive to the ABL-aspect ratio, as shown in Fig. 16 (c). For  $\tilde{H}_i \leq 0.167$ , the power profile follows the trend as that of Fr=0.15 and  $Fr_i=1.3$  because these are the constant values simulated while varying  $\tilde{H}_i$ . However, the power profile for the shallowest ABL ( $\tilde{H}_i=0.042$ ) is most affected by localized capping inversion displacement, which can be revisited in Fig. 10. The wind farm blockage is highest for  $\tilde{H}_i=0.042$  and reduces as  $\tilde{H}_i$  increases until it becomes almost independent for  $\tilde{H}_i \geq 0.21$  because the deflected flow at the wind farm entrance barely perturbs the capping inversion. The simulated power of the individual turbines increases with increasing  $\tilde{H}_i$  as inflow turbulence intensity increases, as the ABL gets deeper (Brooke et al., 2023). The simulated power output profiles for  $\tilde{H}_i \geq 0.167$  are almost the same, as the capping inversion hardly affects the flow build-up through the wind farm. Thus, the flow around the wind farm becomes similar to that of a truly neutral boundary layer with the same ABL height. We can infer that the internal boundary layer has sufficient space to grow vertically, and pressure variations caused by stratification effects or gravity waves become smaller with increasing  $\tilde{H}_i$ . For shallow ABLs, the pressure perturbations caused by the stratification effects dominate the internal boundary layer growth and hence the power output profile. Thus, the capping inversion's height determines which phenomenon dominates and, in turn, the wind farm flow aerodynamics. When it is low, the stratification effects are increasingly critical to wind farm performance; otherwise, these effects are secondary when compared to wake losses. Thus, inflow characteristics and wake effects still remain primary to wind farm performance.

The vertical wind farm aspect ratio impacts the row-wise power profile mainly in the first half of the wind farm, as shown in Fig. 16d. Again, fixed values of non-dimensional parameters simulated include Fr=0.15,  $Fr_i=1.3$ , and  $\tilde{H}_i=0.084$ . Thus, the power profiles follow the trend for Fr=0.15 and  $Fr_i=1.3$ , except that the highest power-producing row among the waked turbines shifts from the fifth row to the fourth as  $S_h$  increases. Higher velocities for increasing  $S_h$  are a clear factor in improved wind turbine and farm powers. However, stratification effects, though small, are still relevant as increasing  $S_h$  means putting the turbines closer to the capping inversion, which is similar to stronger perturbations and a different response for varying  $S_h$ .

Altogether, it is obvious that  $\tilde{H}_i$  is the most critical physical non-dimensional parameter in terms of wind farm performance. Therefore, for low  $\tilde{H}_i$ , Fr and  $Fr_i$  can alter the capping inversion shape characteristic of the underlying  $\tilde{H}_i$ , thus affecting the row-wise power output.  $S_h$  can affect the power output mainly because of placing the wind turbines at higher or lower wind speeds, which has little to do with gravity waves or blockage. While simulating CNBLs,  $\tilde{H}_i$  becomes more critical for shallow ABLs and will be of even greater importance for stable surface conditions with shallower ABLs than in CNBLs.

Preprint. Discussion started: 19 November 2025

© Author(s) 2025. CC BY 4.0 License.




#### 5 Conclusions

This study provides a systematic investigation of wind farm-atmosphere interactions under neutral surface conditions using LES. Among the tested simulation configurations, CNBL simulation configurations that fully resolve AGWs are likely the most realistic for capturing stratification effects and differ significantly from TNBL cases. The CNBL configurations that only partially resolve AGWs predicted similar power outputs to rigid-lid cases, which are less practical. We showed that capping inversion displacement plays a central role in coupling the fluctuations in the ABL with those in the free atmosphere (i.e. AGWs). Interfacial and internal waves were identified as the main AGW types shaping the capping inversion, such that interfacial wavelength and the formation of converging and diverging zones in the ABL are correlated. The trapped waves confined within the capping inversion were transient and negligible. Overall, AGW behavior depends strongly on the ABL aspect ratio ( $\tilde{H}_i$ ) and the Froude numbers of the free atmosphere (Fr) and the capping inversion ( $Fr_i$ ). The interfacial wavelength reduces for decreasing  $Fr_i$  and approaches a quarter of the wind farm length for  $Fr_i = 0.25$ . Whereas the amplitudes grow stronger for shallower inversions. Internal wave characteristics depend mainly on Fr.

Wind farm performance was most sensitive to the ABL aspect ratio: shallow inversions increased wind farm blockage and reduced wind farm efficiency, while deeper layers facilitated wake recovery and improved power output. Increasing Fr,  $Fr_i$ , and the wind farm aspect ratio  $(S_h)$  reduced blockage, enhancing efficiency. Localized spatial variations in power arose from the converging and diverging zones in the ABL associated with the wavy capping inversion. As a result, the total wind farm efficiency—which accounts for both wake losses and stratification effects—remained nearly constant with respect to Fr and  $Fr_i$ , but improved with increasing  $\tilde{H}_i$  and  $S_h$ . These findings suggest that increased wind farm blockage always happens when there is stratification, primarily affecting the front row of turbines. Although weaker than the blockage effect, stratification-induced flow variations can cause downstream turbines to gain or lose power, but these effects were often dominated by stronger wakes.

This work clarifies the role of AGWs, which exist only in the stably stratified atmosphere, in wind farm–atmosphere interactions and establishes the importance of selecting appropriate LES configurations. Beyond providing a coherent description of wind-farm-induced AGWs, it also gives guidelines for expected AGW characteristics such that one can configure reflection-free domains in CFD simulations. Although this study spans a wide range of CNBL conditions, it remains largely confined to canonical CNBL setups. We use a simple actuator disc model that is sufficient for generating gross wake effects and for estimating relative power levels. Therefore, we present relative trends in wind farm performance, as opposed to absolute changes, as a function of changes in stratification. Future extensions could explore canonical stable and unstable ABLs, realistic atmospheric conditions, and more advanced actuator models. The results also point to the need for further study of clustered wind farms, particularly under shallow boundary layers where AGWs (i.e. stratification) have clear impacts.

Data availability. The raw data from LES and post-processing scripts will be archived as a repository on the 4TU drive of TU Delft and will be open access.

https://doi.org/10.5194/wes-2025-236 Preprint. Discussion started: 19 November 2025

© Author(s) 2025. CC BY 4.0 License.

Video supplement. Videos can be made available on request.

Author contributions. Conceptualization, M.A.K, M.J.C, S.J.C; methodology, M.A.K, M.J.C; software, M.A.K; validation, M.A.K; formal analysis, M.A.K; investigation, M.A.K; computational resources, M.A.K; data curation, M.A.K; writing–original draft preparation, M.A.K; writing–review and editing, S.J.W, M.J.C.; visualization, M.A.K; supervision, S.J.W, M.J.C; project administration, M.A.K, S.J.W; funding acquisition, S.J.W. All authors have read and agreed to the published version of the manuscript.

Competing interests. No competing interests are present.

*Acknowledgements.* This publication is part of the project: Numerical modelling of Regional-Scale Wind Farm Flow Dynamics, with project number: 2023/ENW/01454045 of the research programme: ENW which is (partly) financed by the Dutch Research Council (NWO).

R

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

© Author(s) 2025. CC BY 4.0 License.

730

- Stevens, R. J., Graham, J., and Meneveau, C.: A concurrent precursor inflow method for Large Eddy Simulations and applications to finite length wind farms, Renewable Energy, 68, 46–50, https://doi.org/10.1016/j.renene.2014.01.024, 2014.
- Stieren, A. and Stevens, R. J.: Impact of wind farm wakes on flow structures in and around downstream wind farms, Flow, 2, E21, https://doi.org/10.1017/flo.2022.15, 2022.
  - Stipa, S., Ajay, A., Allaerts, D., and Brinkerhoff, J.: TOSCA An Open-Source Finite-Volume LES Environment for Wind Farm Flows, Wind Energy Science, 2023.
  - Stipa, S., Ahmed Khan, M., Allaerts, D., and Brinkerhoff, J.: A large-eddy simulation (LES) model for wind-farm-induced atmospheric gravity wave effects inside conventionally neutral boundary layers, Wind Energy Science, 9, 1647–1668, https://doi.org/10.5194/wes-9-1647-2024, 2024.
  - Taylor, J. R. and Sarkar, S.: Internal gravity waves generated by a turbulent bottom Ekman layer, Journal of Fluid Mechanics, 590, 331–354, https://doi.org/10.1017/S0022112007008087, 2007.
  - Wu, Y.-T. and Porté-Agel, F.: Modeling turbine wakes and power losses within a wind farm using LES: An application to the Horns Rev offshore wind farm, Renewable Energy, 75, 945–955, https://doi.org/https://doi.org/10.1016/j.renene.2014.06.019, 2015.