# Peer review of "Dependence of Wind-Farm-Induced Gravity Waves and Wind-Farm Performance on Non-Dimensional Atmospheric Parameters and Simulation Configuration"

_Wind Energy Science, 2025_

## Referee Comment (RC2)

**Review report**

**Title:** Dependence of Wind-Farm-Induced Gravity Waves and Wind-Farm Performance on Non-Dimensional Atmospheric Parameters and Simulation Configuration

**Authors:** Mehtab Ahmed Khan, Matthew J. Churchfield, and Simon J. Watson

The introduction of this paper states a "need to describe the AGW phenomena clearly, distinguish all the kinds of AGWs, and identify how each kind influences wind farm performance and simulation set-up." This well-written manuscript largely achieves these aims. In addition, the systematic variation of relevant non-dimensional parameters in this numerical study provides insight into how various inflow and wind farm conditions influence wind farm flows. That said, I see a number of minor issues in the manuscript and one major issue that should be addressed before publication. I will start with the minor issues and finish with the one major issue.

**Minor issues**:

Line 56: There should be a space between "turbines" and "(Smith, 2023)".

Lines 87-88: I struggled a bit with the "dimensional" vs "non-dimensional" classification of various papers. First a minor point: the "and" in "either dimensional and non-dimensional quantities" should probably be an "or". Plotting the results against non-dimensional quantities is an important part of Lanzilao and Meyers (2024), though I think it is fair to say that dimensional quantities were the quantities specifically varied in the parametric study. Finally, for examples of papers focusing on non-dimensional quantities, should Allaerts and Meyers (2019) be included? Allaerts and Meyers. Sensitivity and feedback of wind-farm-induced gravity waves. JFM, 2019.

Line 155: "...with a thrust coefficient, with normalizing velocity taken at the disk, $C_t'$ of 1.3." This just doesn't read well. Consider re-arranging the sentence.

Line 194: "$H_i$ is capping inversion mid height". Two things here. Firstly, there should probably be a "the" between "is" and "capping". Secondly, could you comment in your response on using the mid-height for the Froude number (a choice I support) relative to line 110, which reports on two papers that "describe interfacial waves as two-dimensional waves somehow bound to the top of the capping inversion"?

Line 225: The "s" in "Ls" should be a subscript.

Line 310: "AGWS" should probably be "AGWs"

Line 336: To my ear, "constraining" would probably be better than "to constrain".

Line 376: For clarity, "turbine" should probably be replaced with "row" or "row of turbines". Same is true for line 377.

Line 384: When I hear "power curve", my mind goes immediately to the power vs freestream wind speed curve commonly used to define the power performance of an individual turbine. However, in the context of this paragraph, a power curve clearly represents something else (row-by-row variation in temporal and row-averaged power). I suspect that the "power curve" labels could also make other readers pause. You may wish to consider writing something like "row-by-row power output" instead.

Lines 406-409: This text focuses on Figure 7 and whether and to what degree there are spurious AGW reflections. In a couple of places, the text highlights that these reflections can be seen in Figure 7. Is there anything the authors can do to help the reader understand the features of these plots that signal a reflection? I do not doubt that the text is correct about reflections occurring in these two cases (top and bottom plots in Figure 7). I just do not trust that I am able to positively identify the telltale signs of reflections in these plots.

Figure 6b. The variation in power production between the five cases is small. (The case-to-case power variation appears to be much less than in Figure 16a and Figure 16b, where the authors see little variation in the wind farm efficiency.) It is almost as if practically speaking the damping strength doesn't matter in this case. Do you think this is generally true or is there something about this case that makes the turbine power predictions generally insensitive to the damping strength.

Line 464: "In a physical sense, $Fr_i$ approaching 0 would mimic a rigid lid that would barely move due to the disturbance from a wind farm." I think this line might be OK, but I'm not sure. In this study, $Fr_i$ is varied by adjusting delta theta (change in potential temperature across the inversion). If we drive $Fr_i$ in the direction of zero by strengthening the inversion in any sort of realistic way, aren't we creating a situation that is more like free surface with a fluid of uniform density below?

Lines 449-500: Would replacing "calculated" with "theoretical" and "measured" with "simulated" help avoid confusion?

Line 507: It is not immediately obvious what is meant by the capping inversion becoming "more sensitive" for sub-critical values of $Fr_i$. I believe the authors are saying that the displacement of the inversion layer due to the presence of the wind farm is more pronounced. Instead of "the capping inversion becomes more sensitive", it might be better to write something more specific and less open to interpretation.

Line 509: "... showed that lambda approaches the streamwise turbine distancing as if each turbine row is triggering an interfacial wave." I *think* I know what the authors are trying to say here, but I'm not sure. Perhaps with some small tweaks readers may be more likely to get your point quicker. For example, "... showed that lambda is on the order of the distance between turbine rows – as if each row is triggering an interfacial wave."

Lines 526-527: I had to go back to Lanzilao and Meyers (2024) to decipher what this sentence is referring to. I believe the sentence is referring to the orientation of the waves along the inversion north and south of the wind farm when viewed from above. Lanzilao and Meyers call it a V-shaped pattern. Now after checking the Lanzilao and Meyers (2024) the meaning of this sentence seems clear, but for the reader who does not have a picture in their head of what this refers to, more description may help. Perhaps the word "inclination" set me down the wrong mental path, as I typically think of an inclination being in a vertical plane.

Lines 556-557: "For instance, the lamda/L plot against $Fr_i$ resembles the decreasing inclination angle trend against $Fr_i$ shown in Fig. 13." I believe the lamda/L plot against $Fr_i$ refers to the top right plot in Figure 14. It is not clear to me how this plot resembles the top right plot in Figure 13. Am I missing something here?

Line 596: To avoid confusion, I recommend replacing "a higher capping inversion" with "an upwards displacement of the capping inversion."

Line 619-620: "However, the power profile for the shallowest ABL ($H_i = 0.042$) is most affected by localized capping inversion displacement, which can be revisited in Fig. 10." Figure 10 shows *maximum* capping inversion displacement, but the sentence is talking about *localized* capping inversion displacement. Should the reader then be directed to look at Figure 9?

Lines 654-655: "Whereas the amplitudes grow stronger for shallower inversions." This is a dependent clause. It should not stand alone as a complete sentence. That said, I think it is OK to break grammatical rules under certain circumstances. The text is clear enough. I just wanted to point this out in case this was done accidentally rather than on purpose. There is another issue with this line: the term "shallower inversions". I am pretty sure this refers to inversions that are closer to ground. Up until this point in the manuscript, this situation was described as a shallower ABL. When I first read this, I thought "shallower inversion" might refer to the thickness of the inversion, but of course, inversion thickness is not investigated in this study, so it almost certainly refers to ABL thickness/depth. "Shallow inversions" come up again in the next paragraph. I recommend switching back to shallow boundary layers or shallow ABLs for consistency and to avoid confusion.

**Major issue:**

The one major concern has to do with analyses and conclusions related to wind farm "performance" and "efficiency". Figure 16 shows time-and-row-averaged power vs. row for four different parametric studies. These plots seem to be used in support statements in the paper related the impact of the parameters on wind farm performance and efficiency. I think that "performance" and "efficiency" mean the same thing in this paper, but I am not 100% sure. These terms should be specifically defined.

Wind farm efficiency is traditionally defined as the power of the wind farm divided by the sum of the power the turbines would produce if they were operating in isolation. If turbines were run in isolation for each of the simulated conditions, I recommend normalizing the power values in figures 6 and 16 with the isolated power values. Simulating each wind farm turbine in isolation for each set of inflow conditions is not practical; the simulation of just one of the wind farm turbines in isolation may be enough if the time-mean flow is sufficiently horizontally homogeneous.

If isolated turbine simulation results are not available and producing the results deemed to be too taxing, an alternative power normalization representing a relevant reference power of some sort for each case could still be useful. The authors acknowledge this general issue on Line 641: "$S_h$ can affect the power output mainly because of placing the wind turbines at higher or lower wind speeds, which has little to do with gravity waves or blockage." This is no doubt true. What about the other sensitivity studies? Would the power of an isolated turbine be the same across all the $H_i$ cases, for example? When $H_i$ varies, the shear and veer will probably vary, which could change the power across different cases for a given hub-height wind speed. (The inflow turbulence will probably vary as well, which may affect wake recovery.)

There are two key statements in the Conclusions section about the impact of these various non-dimensional parameters on wind farm efficiency (there are also some in the last paragraph of the abstract):

Statement 1: "Wind farm performance was most sensitive to the ABL aspect ratio: shallow inversions increased wind farm blockage and reduced wind farm efficiency..." In addition to defining wind farm efficiency, the definition of wind farm blockage should be clarified. At the least, the reader needs to understand how the authors reached the conclusion that shallow boundary layers increased wind farm blockage. Is wind farm blockage the amount by which blockage reduces production in the leading row? If so, how is the reduction quantified? Relative to isolated operation? I suspect the wind farm blockage claim is based on the power variation across the cases in the first row. Since the freestream, hub-height wind speed is the same across these cases, one might attribute the power variation to wind farm blockage. I suspect this, in fact, is the dominant factor explaining the power variation in the first row, but based on the information in the paper, I cannot rule out another contribution: different inflow conditions could result in different powers for turbines simulated in isolation (power variation unrelated to wind farm blockage), as the power at the actuator disks is based on the velocity across the disk rather than the hub-height freestream wind speed.

Statement 2: "... the total wind farm efficiency—which accounts for both wake losses and stratification effects—remained nearly constant with respect to Fr and $Fr_i$, but improved with increasing $H_i$, and $S_h$." Again, how was wind farm efficiency in these cases determined? It is not clear from the text or the figures, especially for the $S_h$

sensitivity plot in Figure 16. In fact, I think the only places efficiency is mentioned are in the abstract and in the conclusion.

On a related note, the abstract claims that deeper layers enhance wake recovery. I think that is true, but I didn't see this specifically demonstrated and called out in the paper. How specifically is the amount of wake recovery determined?

To summarize this concern, it is not clear how the conclusions regarding wind farm performance, wind farm blockage, and wind farm efficiency are supported (or even how these metrics are defined). I suspect that the conclusions are based on Figure 16, but from these plots it is difficult for me to discern case-to-case power variation due to wind farm effects from case-to-case power variation due to differences in the wind conditions that an isolated turbine would experience (particularly for the sensitivity studies where $H_i$ and $S_h$ are varied). Lastly, if efficiency/performance trends remain a key point of discussion in the paper, I recommend quantifying the wind farm efficiency for each case (same for front-row blockage loss if it is discussed). Even if the row-by-row data in Figure 16 were normalized by isolated power, for example, it may not be straightforward for someone to mentally translate the plots into reliable efficiency/loss trends for the full wind farm.